# Detection performance of an X-band marine radar system for free-flying Asian particolored bats (*Vespertilio sinensis*)

Takahiro Sato[1,2☯], Yoichi Kawaguchi [1,2☯]*

1 Graduate School of Technology, Industrial, and Social Sciences, Tokushima University, Tokushima, Japan, 2 Sado Island Center for Ecological Sustainability, Niigata University, Sado, Niigata, Japan

☯ These authors contributed equally to this work.
* kawaguchi@ce.tokushima-u.ac.jp

## Abstract

The rapidly growing number of wind farms is an increasing threat to many bat species and populations worldwide. Monitoring and prediction of bat collision risk requires techniques capable of quantifying bat movements over large spatial areas. Marine radar systems are highly effective tools for observing flying animals over large surveillance areas. However, the detection capability of marine radar systems has rarely been validated for bats. In this study, we sought to validate the effectiveness of a commercially available X-band marine radar in detecting Asian particolored bats (*Vespertilio sinensis*). To achieve this goal, we first examined the effect of adjusting the height of the radar antenna to reduce ground clutter. By lowering the antenna height from 2.2 m to 0.4 m, the visible area on radar images was increased by a factor of 2.1 over the entire 1.5 km observation range. Subsequently, we conducted radar observations under conditions where radar-detected targets could be reliably identified as bats. To acquire ground truth bat flight trajectories, we tracked bats at dusk using an ornithodolite simultaneously with radar scans. We manually tracked bat echoes on radar images and matched them to ornithodolite trajectories. Following this procedure, we identified 25 radar tracks as bat flights. These tracks were analyzed to determine the probability of echo detection (detection versus non-detection) as a function of distance from the radar by applying a generalized linear mixed model. The model analysis demonstrated that X-band marine radar was capable of detecting bat flights at a distance of up to 1.0 km, with a probability of detection exceeding 70%. Our results suggest that X-band marine radar can be an appropriate tool for monitoring bat movements, and also contribute to the establishment of range settings that should be considered for radar bat surveys.

**Data availability statement:** All relevant data are within the manuscript and its Supporting information files. Additionally, raw data files are available from the Tokushima University Institutional Repository (https://tokushima-u. repo.nii.ac.jp/records/2013473, accession number https://doi.org/10.15000/0002013473).

**Funding:** This study was supported by the Environment Research and Technology Development Fund (JPMEERF20214G02) of the Environmental Restoration and Conservation Agency (ERCA, https://www.erca.go.jp/), provided by the Ministry of Environment of Japan to YK, and Japan Society for the Promotion of Science (JSPS, https://www.jsps.go.jp/) KAKENHI Grant Numbers JP24K23923 (Grant-in-Aid for Young Scientists (Start-up)) to TS, and JP23K18545 (Grant-in-Aid for Challenging Research (Exploratory)) to YK. The funders had no role in the study design, data collection and analysis, decision to publish, or preparation of the manuscript.

**Competing interests:** The authors have declared that no competing interests exist.

## Introduction

Since 2000, wind turbine collision fatalities are the leading cause of multiple bat mortality events worldwide [1–4]. In North America, the estimated annual mortality rate for bats ranged upwards to hundreds of thousands of bats per year [3,5]. A comprehensive analysis of cumulative collision fatalities involving bats in 20 European countries over the period 2003–2014 revealed a total of 5,815 fatalities, affecting 27 distinct species of bats [6]. Furthermore, bat mortality at wind farms has also been observed in Asia, such as Taiwan, which has recently implemented measures to promote wind power generation [7]. Further negative impacts may include habitat loss, alterations in habitat utilization patterns, and a threat to the viability of bat populations [8,9].

The scientific study of flight behavior and spatiotemporal movements of bats is crucial for monitoring and predicting collision risk in environmental impact assessments. However, detecting bats in flight has been a challenge due to their diminutive size and exceptional mobility. Acoustic surveys have been extensively employed as a methodology to investigate the distribution and activity patterns of bats [1,10,11]. Acoustic bat detectors are capable of identifying the presence of species and quantifying their activity using an index known as 'bat pass'. However, bat detectors lack the capability to quantify the number of individuals and have a restricted detection range for bat calls (approximately 50 meters) [12,13]; this represents a disadvantage of acoustic surveys. Thermal infrared imaging cameras have also been utilized to observe bat behavior [14,15], unfortunately the field of view has to be narrowed in order to detect distant targets. Bat monitoring necessitates a technique capable of generating large spatial scale data on bat movements.

Radar technology is a powerful tool for detecting flying animals over large spatial areas. Marine radar systems have been widely used in ornithological studies since the 1950s to investigate both large-scale migrations and local-scale movement patterns of birds [16–20]. More recently, dedicated avian radars (e.g., DeTect, USA; Robin Radar Systems, the Netherlands) have also been employed to monitor avian collision risk at wind farms and airports [21,22]. At shorter ranges (~ 6 km), marine radars have the advantage of tracking individual targets and quantifying the number of flights [1,20]. The utilization of marine radar for the study of bats holds the potential to enable the detection and tracking of bat movements over large spatial areas [23,24]. However, only a limited number of studies have been conducted to investigate bat movements by marine radars [14,24–27].

To effectively integrate marine radar into bat research, it is crucial to comprehend its capabilities and limitations [28–31]. Received signal power from targets decreases exponentially with distance. This results in reduced detectability as targets move further away from the radar. Previous radar studies have indicated that marine radars possess the capability to detect bats at a distance of approximately 1.5 km [25–27]. However, they have not examined variations in detection probability over distance. Kreutzfeldt et al. [23] has simulated the received signal power of bat-sized objects over distance using an analytical volume model. Although their approach was

practical for radar calibration, the detectable range was determined using an artificial bat-sized model in stationary conditions, not in-flight. The effect of distance on detection probability should be investigated for free-flying bats to enhance our understanding of the performance of marine radar for quantitative bat surveys. Since commercial off-the-shelf marine radars are not specifically designed to detect flying animals, detection performance also depends on system settings. Furthermore, the landscape around the radar is also a limiting factor in target detection and tracking [19,30]. Depending on the installation location, reflections, from objects such as buildings, vegetation, and the ground surface itself, create clutter zones that reduce the detectable area. Clutter reduction techniques need to be applied according to the environmental conditions of each radar installation site.

Accurate target identification is a major challenge when detecting flying animals using radar. Differentiating between nocturnal targets (birds, bats, or insects) detected by radar has often relied on flight speed and/or behavior [25,26]. Nocturnal studies have also utilized additional techniques, such as acoustic detectors, thermal imaging, or night vision cameras, to gather data confirming the presence of bats within the radar coverage area [14,32]. However, it remains a significant challenge to directly align radar targets with other data, such as acoustic bat passes, that can substantiate bat flights [24]. Contrary to the previous approaches, recent research using an ornithodolite has successfully matched tracks extracted from radar images to avian flight trajectories [33]. This technique could be beneficial in obtaining ground truth data for the purpose of verifying the radar detection performance.

The purpose of this study is to verify the performance of an off-the-shelf marine radar system for detecting bat flights. To achieve this objective, the first step was to identify an installation method specifically designed to minimize ground clutter, which significantly impacts the detectable area. Subsequently, the movement trajectories of bats were captured using an ornithodolite to align them with tracks extracted from radar images. This alignment facilitated an analysis of the influence of distance on the detection probability.

## Methods

### Study site and species

Our field work was conducted at a single location (37°46.85'N, 138°58.75'E, 0 m above sea level) in a rice field area of Niigata, Niigata Prefecture. We placed a radar unit in a small open space surrounded by rice fields (Fig 1a). The landscape within the radar coverage area was comprised of rice fields, ditches, roads, residential areas, and a viaduct. We focused observations of bat flights on the southern area of the radar site with flat terrain and few tall structures blocking the radar beam (Fig 1b). Our field work at this site was conducted with the permission of the land manager, the Land Improvement District of Nishikanbara (permit number 486).

In this study, we focused on the Asian particolored bat (*Vespertilio sinensis*, weighing 14–30 g) to verify radar detection performance. In eastern Japan, this species is known for its extensive migratory patterns, which unfortunately often result in its demise due to collisions with wind turbines [34]. Since 2021, we have surveyed bat roosts in Niigata City every summer (June–August), including the area of this study. To date, a large number of maternity colonies of Asian particolored bats have been found along the viaduct (Fig 1a) running through the study area. Within 1.5 km of the radar, we identified a total of 67 roosts and at least 1,000 individuals. As bats emerged from their roosts shortly after sunset, their flights could be observed directly from the rice fields adjacent to the viaduct.

As preliminary surveys, we also made stationary visual observations to identify species appearing in the target area at dusk (according to the observation area shown in Fig 1b). Observations were conducted for 16 days in 2022 (July and August) and two days in 2023 (early July), at times from five minutes before sunset to 30 minutes after sunset (total 35 minutes). For each day, a single observer recorded the species crossing an observation site located within the radar coverage area. An acoustic bat detector (Echo Meter Touch 2 Pro, Wildlife Acoustics, Maynard, MA, USA) was used for species identification. The device can detect low-frequency (15–30 kHz) bat echolocation calls up to approximately 40 m

## (a) Study site

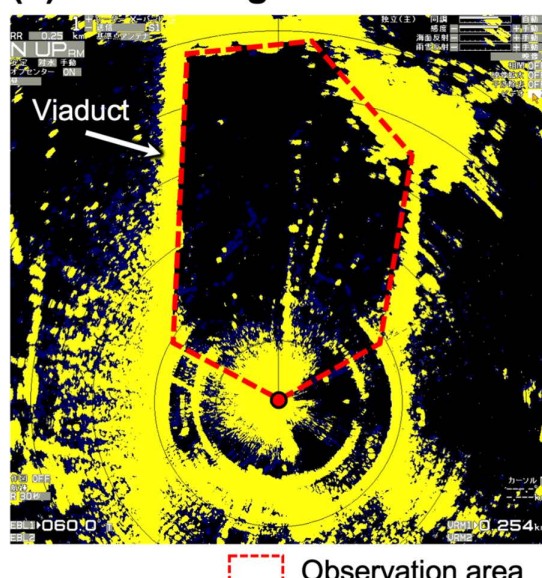

**Viaduct**

**Radar**

**500 m**

- Paddy field
- Cropland
- Grassland
- Man-made structure
- Greenhouse
- Trees

## (b) Radar image

**Viaduct**

⬚ Observation area

**Fig 1. Study site and observation area. (a)** shows a map of the study site. Roosts of Asian particolored bats are located along the viaduct. The white square corresponds to a radar coverage area. Basemap reprinted from High-Resolution Land Use and Land Cover Map (Japan) Version 25.04, under a CC BY license, with permission from Earth Observation Research Center (EORC), The Japan Aerospace Exploration Agency (JAXA) (dated 17 July 2025), original copyright 2025. In **(b)**, a radar observation area for detecting and tracking bat echoes is shown by the red polygon. The red dot shows the radar position.

from the microphone. We were able to identify calls of Asian particolored bats in the field based on call type (frequency modulated/quasi constant frequency call) and frequency (25 kHz) [35]. Throughout the observations, only Asian particolored bats were detected. When birds appeared, we also identified the species using 8 × 42 binoculars (Monarch 7, Nikon Corporation, Tokyo, Japan) with an 8° field of view. During the bat observations, only a few bird species were observed. The identified birds consisted of two species of heron (grey heron *Ardea cinerea* and great white egret *A. alba*) and carrion crow (*Corvus corone*).

## Radar system

We used a commercially available X-band marine radar MDC-7910 (Koden Electronics Co., Ltd., Tokyo, Japan). The MDC-7910 consists of a scanner unit with a magnetron-based amplifier (Koden RB718A; 12 kW output power) and a horizontally rotating slotted waveguide antenna (Koden RW701A-06; 1.95 m length) operating at a speed of 48 rpm. Radar images (1280×1024 pixels) were captured from the radar display unit to a laptop computer, which recorded images to a hard disk drive at 1.3-s intervals corresponding to the antenna rotation speed. The radar image has a resolution of 1.6 m per pixel in both XY directions when the observation range is set to 1.0 km. An overview of the radar system specifications and settings is provided (see Table 1). The use of the marine radar for field research was approved by the Experimental Radio Station of the Ministry of Internal Affairs and Communications, Japan (Permit Number: Shikoku Bureau Telecommunications #588).

We adjusted the gain manually so that a small number of noise blips (single pixels) just started to appear on the display. The MDC-7910 radar has 15 levels of echo signal resolution, and RGB (red, green, blue) values can be set for each level. Using the manufacturer's default settings, echo colors on the lower side of the levels are invisible on the display (S1 Fig). To increase the visibility of moving targets with weak signal intensity, we set to the same color (R: 255, G: 255, B: 0) for all 15 levels (S1 Fig). The range for the radar was set to 1.0 km when observing bat flights. Radar surveys were performed for a total of five days in early July 2023 under conditions with no rainfall or fog, both of which affect radar detection performance. Based on the preliminary surveys described above, we only conducted radar observations while large numbers of Asian particolored bats were emerging from their roosts and passing through the radar coverage.

## Ground clutter reduction

Ground clutter creates invisible zones and makes it impossible to detect small moving targets when scanning the horizontal plane. Although the appearance of ground clutter is inevitable, objects surrounding the radar, such as low vegetation, hills, and man-made structures, can be used to reduce ground clutter [1,19]. Before starting bat observations, we determined an optimal radar installation method for our study site to reduce the effect of ground clutter. In the study area, the vegetation height of the rice fields was 30–40 cm higher than the ground level. We expected that matching the height of the radar antenna to the height of the vegetation would reduce ground clutter. The reason was that the nearest vegetation from the antenna was expected to shield the lower half of the emitted radar beam, as mentioned in the existing literature [16]. To substantiate this prediction, we quantified an area of ground clutter on radar images at two antenna heights: 0.4 m and 2.2 m (Fig 2). The antenna unit was placed 5 m away from the nearest rice field. For the ground clutter measurement, the radar range was set to 1.5 km.

We used ImageJ software (US National Institute of Health, Bethesda, Maryland, USA) to measure the area of clutter on the radar images. To account for scan-to-scan variations in clutter, we collected ten consecutive images at both

**Table 1. Koden MDC-7910 radar system specifications and settings.**

| Antenna-Scanner unit | Configuration | Processor-Display unit | Configuration |
|---|---|---|---|
| Frequency | 9410 MHz | Range | 1.0/1.5 km |
| Pulse power | 12 kW | Gain | 75 |
| Pulse length | 0.08 µs | Sensitivity Time Control | Off |
| Pulse repetition rate | 2500 Hz | Rain | Off |
| Antenna length | 1.95 m | Interference Rejection | Off |
| Antenna gain | 28.5 dB | Video | 2 |
| Horizontal beam width | 1.2° | Display resolution (pixels) | 1280×1024 |
| Vertical beam width | 22° | | |

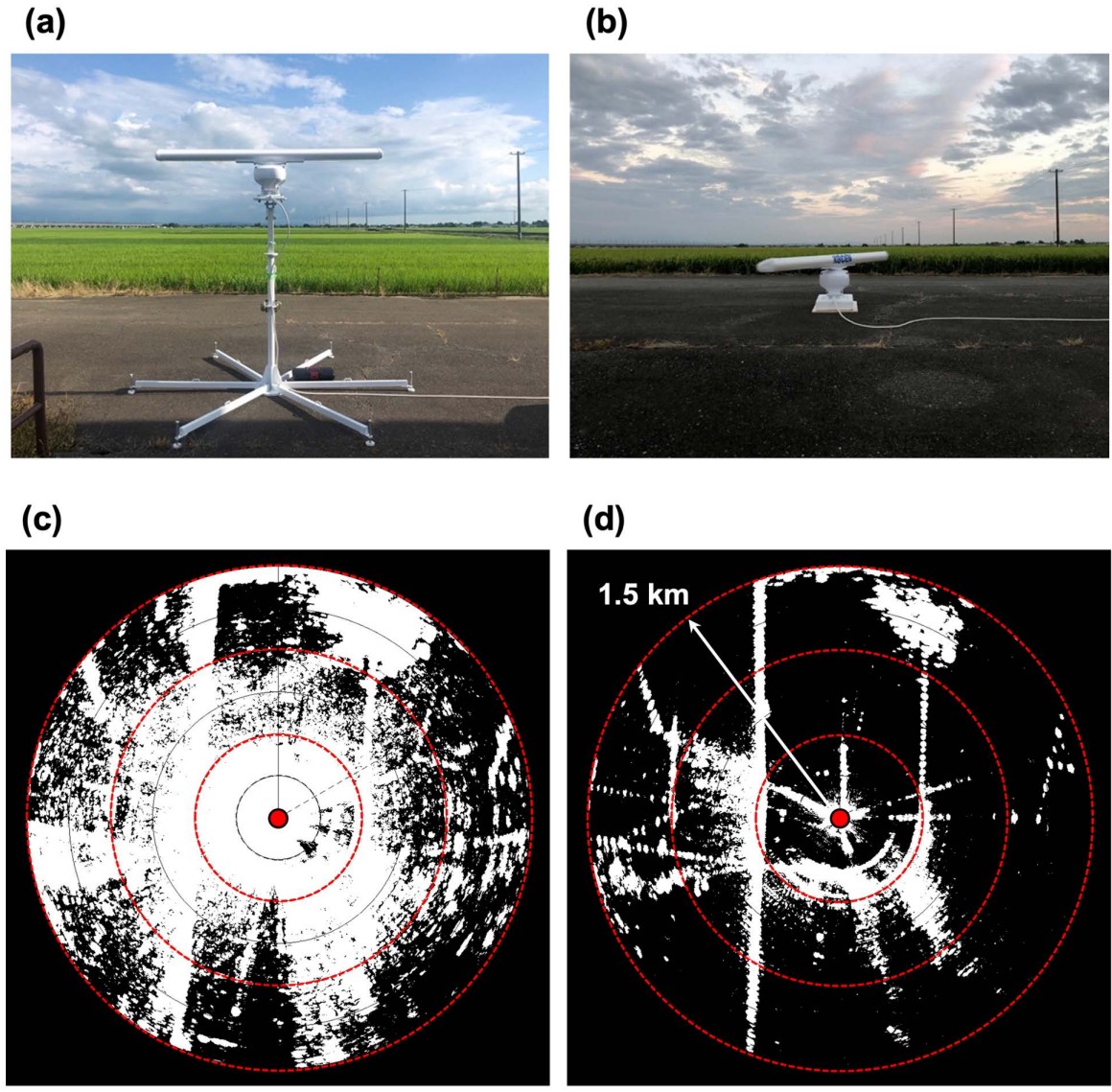

**Fig 2. Radar antenna installation for ground clutter reduction.** Photographs of the antenna unit at two different heights are shown in **(a)** and **(b)**, and the corresponding radar images are shown in **(c)** and **(d)**, respectively. Red circles in **(c)** and **(d)** represent range rings at 500 m intervals. The red dot in **(c)** and **(d)** shows the radar position.

antenna heights (2.2 m and 0.4 m). The images were converted to grayscale and a mean brightness value (range: 0–255) was calculated for each pixel. We binarized the images with a brightness threshold of 100 and then calculated the percentage of clutter area within 500 m radius of the radar, 500–1,000 m range and 1,000–1,500 m range. Calculated data is available in S1 Table.

## Ornithodolite tracking

We conducted ornithodolite tracking simultaneously with radar scans (0.4 m antenna height) to collect flight trajectories of bats. We used an ornithodolite (Vector 21 Aero, Safran Vectronix, Heerbrugg, Switzerland, 1σ distance error: ±5 m, 1σ

elevation error: ± 0.2°, 1σ azimuth error: ± 0.6°), which can calculate the 3-D positions of a target by automatically measuring the azimuth, elevation angle, and straight distance. By using an ornithodolite support system LMS 1.0 software (Kyokuto Boeki, Tokyo, Japan), we were able to track bats at 2.3-s intervals. To obtain bat trajectories over the entire range of the radar observation area, three ornithodolite tracking sites were established (Fig 1a). A field observer started observations five minutes before sunset and ceased 40 minutes thereafter. Observations were conducted on 4–7 July and 9 July 2023 at one site per day. The mean distance from the observer to bat positions recorded by the ornithodolite was 356.1 m (range = 112.5–690.1 m). Trajectories obtained with at least three tracking points were utilized to identify bat flights in the radar image analysis described below. Tracking data is available in S2 Table. In this study, no animal handling was conducted, so no ethics permission was required.

### Bat echo extraction

To identify radar tracks associated with bats, we employed a two-step process. 1. Manual Tracking: we manually tracked echoes that exhibited potential bat-like characteristics on radar images. 2. Track Extraction: subsequently, we extracted tracks that closely matched the trajectories of bats obtained by ornithodolite.

We used image processing software Fiji [36] to track echoes. For each day, we extracted radar images during the time of ornithodolite observations. Bird species observed in this area at dusk (grey heron, great white egret, and carrion crow) were generally scarce and only occasionally appeared within the radar observation area. In order to minimize contamination from bird movements, we focused only on echoes that emerged from the area around the locations of bat roosts (viaduct) and moved westward or southwestward. These were the major directions of the movement of bats observed in our study site. We systematically examined images frame-by-frame and identified echoes that gradually shifted their position within the observation area (Fig 3a). Using 'Manual Tracking' implemented within Fiji, we detected the center coordinates of echoes measuring at least three pixels in size (Fig 3b). We excluded flock-like echoes that merged with other echoes or separated into multiple echoes during tracking. When non-detection events occurred (disappearance of tracking echo), the frame was skipped and the subsequent image was examined. In the event of multiple candidate echoes appearing after skipping, to avoid false tracking, we ceased monitoring at the preceding frame. We also discontinued tracking procedure when echoes were lost for more than four consecutive frames due to the difficulty in detecting the same targets. The non-detection coordinates were estimated to be equally spaced according to the number of radar scans between detected echoes. Distances and azimuthal angles between echo centroids and the radar coordinate were calculated for each track (including non-detections). To project tracks onto a map, the geographic coordinates of echoes were calculated using the distances and azimuthal angles. A total of 386 tracks were obtained as potential data belonging to bats. Furthermore, we examined whether the movement speed of the obtained radar tracks overlapped with that of bats rather than birds or insects. For the radar tracks, the mean ground-speed was 10.4 m/s (range = 4.2–16.5 m/s), suggesting that the majority of the tracked echoes were flying vertebrates since the typical flight speed of insects is less than 5 m/s [32,37]. This data was consistent with the ground-speed of Asian particolored bats (mean = 9.9 m/s, range = 5.4–16.4 m/s) recorded by the ornithodolite. For birds observed in the same study area, the mean ground-speeds of grey herons, great white egrets, and carrion crows recorded by the ornithodolite were 25.0 m/s (range = 12.3–37.7 m/s, n = 13), 21.5 m/s (range = 10.6–37.9 m/s, n = 13), and 23.4 m/s (range = 14.4–30.2 m/s, n = 6), respectively. The ground-speed of the radar tracks was not significantly different from that of Asian particolored bats, but was significantly slower than that of birds (Steel-Dwass test: radar tracks vs. bats, W = −2.557, p = 0.348, radar tracks vs. grey herons, W = 8.441, p < 0.001, radar tracks vs. great white egrets, W = 7.943, p < 0.001, radar tracks vs. carrion crows, W = 5.880, p < 0.001). Therefore, we assumed that the manual tracking data were rarely contaminated by birds or insects.

We then extracted radar tracks that overlapped in time, location, and direction of movement with the trajectories obtained by the ornithodolite (Fig 3c, d). We included radar tracks that crossed within a 5 m buffer of the ornithodolite trajectories, taking into account the measurement error of the ornithodolite. After this procedure, twenty-five of the 386

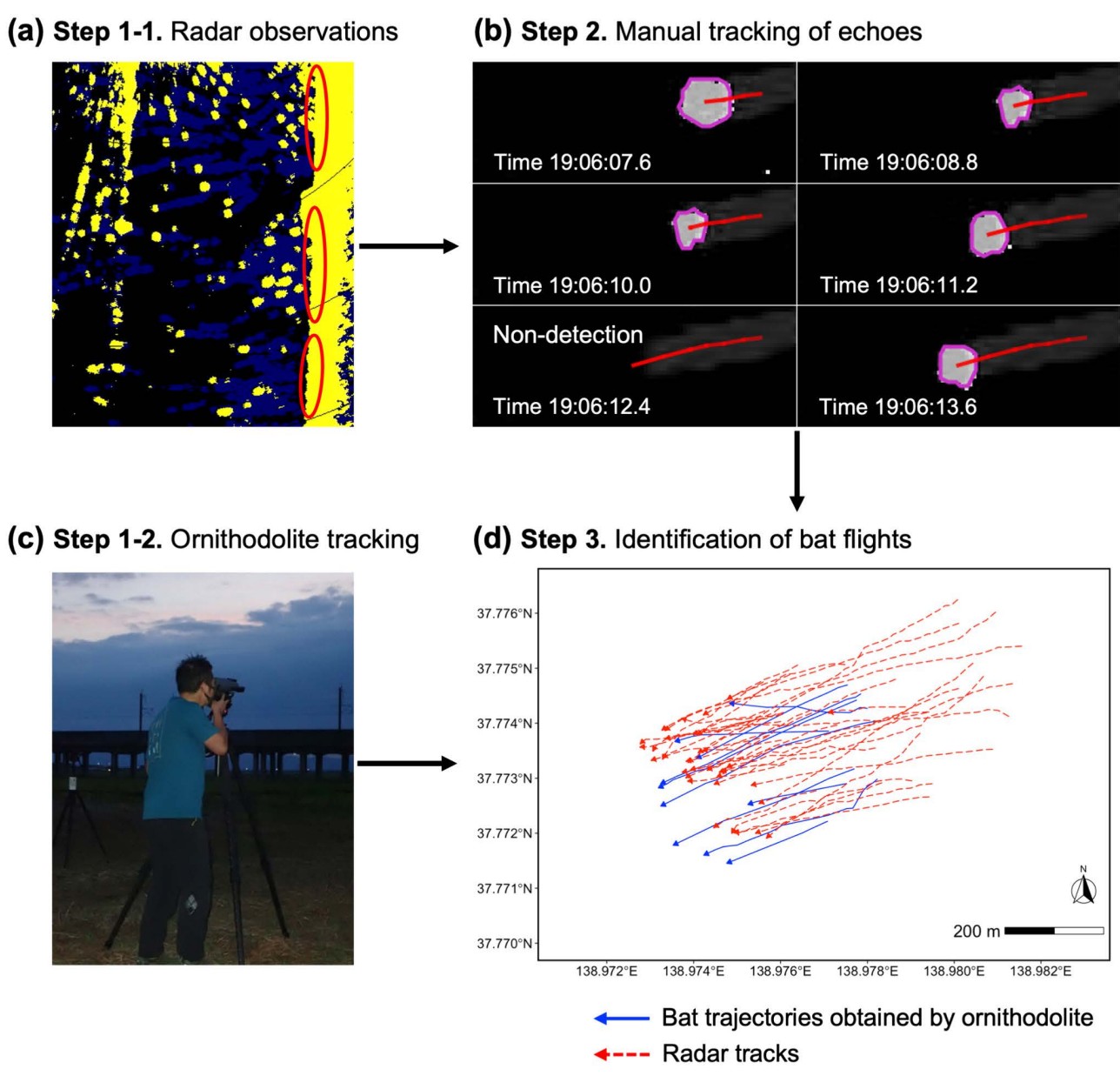

**Fig 3. Procedure of bat flight identification.** Radar observations **(a)** and ornithodolite tracking **(c)** were conducted simultaneously. **(a)** shows the representative radar image of bat echoes emerging from roosts (red circles) at dusk. We examined radar images frame-by-frame and manually tracked echoes that gradually changed position. **(b)** represents a manually tracked echo (blob with pink boundary and red solid line) along six consecutive frames. In this case, the bottom left panel is a non-detection event. **(c)** shows an observer (T. Sato) tracking bats using an ornithodolite. In **(d)**, candidate bat tracks (red dotted lines) extracted from radar images and bat trajectories (blue solid lines) obtained by ornithodolite were projected onto a map (tracks collected on 6 July 2023 are shown).

tracks were identified as bat flights and used as data for statistical analysis (hereafter referred to as "bat tracks"). Manual tracking data is available in S3 and S4 Tables. The individual pictured in Fig 3c has provided written informed consent (as outlined in PLOS consent form) to publish his image.

## Statistical analysis

The purpose of our analysis was to verify the effect of distance on the probability of bat echo detection. However, in addition to distance, bat flight altitude (elevation angle from antenna) also affects radar detectability. We checked for co-linearity between two variables: distance and elevation angle (recorded by the ornithodolite) from the radar position. Since a negative correlation was observed between distance and elevation angle (Spearman's rank correlation: Spearman's rho = −0.71, p < 0.001, S2 Fig), we only used distance in the following model analysis. While orthogonalization or residualization may allow simultaneous analysis for both variables, this approach was not used in this study. This is because it becomes difficult to interpret model estimates after applying these techniques. Additionally, elevation angle alone did not significantly affect the probability of echo detection (S5 Table). To analyze the effect of observation distance on bat echo detection, we used a generalized linear mixed model (GLMM) with a binomial distribution. The model was constructed with detection versus non-detection as the response variable, distance from the radar as the explanatory variable, and track ID as a random factor. We also used a likelihood ratio test to compare the GLMM with a null model (excluding the distance variable). Due to the small number of identified bat tracks (n = 25), we checked the statistical power of the model with the current sample size by conducting a post-hoc power analysis (S1 File). Additionally, we examined how detection probability estimates change with a varying number of included radar tracks. To do so, we modeled the probability of echo detection using subsets of radar tracks filtered by thresholds determined based on the bat's ground-speed (S6 Table). The results showed that the effect of distance on detection probability follows a similar pattern across models (S3 Fig, S6 Table). Although the small sample size and manual selection of radar tracks may affect the generalizability of the model, for this study's GLMM, we used radar tracks extracted by matching bat trajectories recorded by ornithodolite. Statistical analyses were conducted using the software R 4.4.0 [38] with glmmTMB [39] package for GLMM. In the analyses, we assigned significance level at alpha = 0.05.

## Results

The area of clutter on the radar images at an antenna height of 0.4 m was lower than that measured at 2.2 m within a 1.5 km radius of the radar. At the 2.2 m antenna height, the percentage of clutter decreased with increasing distance (0–500 m: 94.9%, 500–1,000 m: 75.0%, 1,000–1,500 m: 43.6%, Fig 4). When the antenna was situated at a height of 0.4 m, the invisible area by clutter was reduced for the entire range (0–500 m: 34.0%, 500–1,000 m: 23.1%, 1,000–1,500 m: 10.8%, Fig 4).

Using the ornithodolite, we obtained a total of 50 bat trajectories in areas ranging from 354.9–1649.6 m (median = 881.3 m) away from the radar. The mean ground-speed of bats was 9.9 m/s (range = 5.4–16.4 m/s), and the mean direction of movement was 250.4° (range = 104.3–305.4°). For the 25 identified bat tracks, the mean ground-speed was 9.7 m/s (range: 6.1 to 13.6 m/s) and the mean movement direction was 253.2° (range = 212.9–289.2°). A total of 382 detections and 75 non-detections were obtained from the bat tracks. The bat tracks consisted of a mean of 15.3 detected echoes (range = 3–41).

Distances between the radar and the bat echoes ranged from 361.7 to 1,048.6 m. The distance showed a weak but statistically significant negative effect on the probability of detection (GLMM: odds ratio = 0.997, 95% confidence interval: 0.995 − 0.999, marginal $R^2$ = 0.053, conditional $R^2$ = 0.053, AIC = 404.1, Table 2). This indicates that the proportion of detection events per track decreases with distance from the radar. Additionally, the model including distance was significantly different from the null model (likelihood ratio test: $\chi^2$ = 7.37, df = 1, p = 0.006). The probability of detection decreased gradually with distance and reached 73% at 1,048.6 m (Fig 5), the maximum distance recorded in this study.

## Discussion

### Bat detection performance

Until recently, there has been limited research conducted to assess the radar detection performance for bats. Furthermore, there is a lack of data to establish an operational range for studying bat movements utilizing marine radar systems [cf. 23, 24]. In this study, we demonstrated the relationship between echo detection probability and distance by conducting

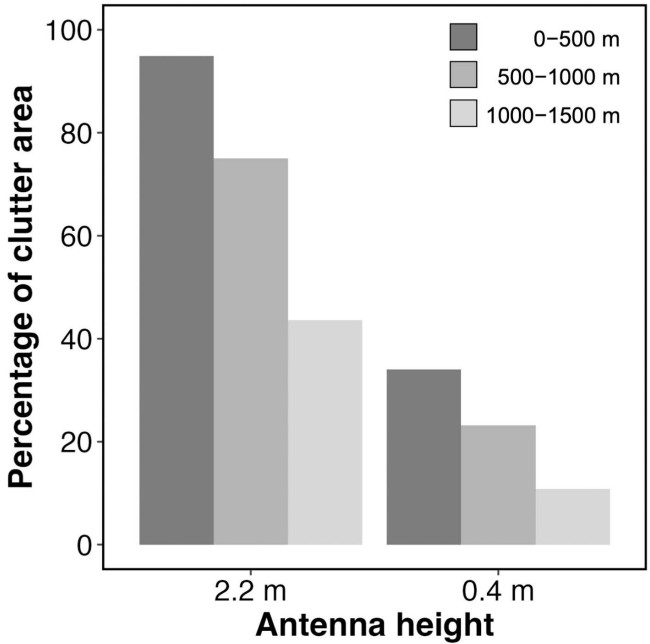

**Fig 4. Reduction in clutter area by adjusting the antenna height.** The percentage of clutter area is shown at each 500 m range interval for two different antenna heights.

**Table 2. The relationship between bat echo detection and distance from radar.**

| Fixed effects | Estimate | Std. Error | Odds ratio | 95% CI | z value | Pr (>| z \|) |
|---|---|---|---|---|---|---|
| Intercept | 3.807 | 0.791 | | | 4.815 | < 0.001 |
| Distance | −0.003 | 0.001 | 0.997 | 0.995−0.999 | −2.870 | 0.004 |
| **Random** | **Name** | **Variance** | **Std.dev.** | **No. of obs.** | **Groups: Track ID** | |
| Track ID | (Intercept) | < 0.001 | < 0.001 | 457 | 25 | |

A generalized linear mixed model (GLMM) consisted of bat echo detection (detection/non-detection) as the dependent variable, distance from radar as the explanatory variable and track ID as a random factor.

radar observations in situations where radar-detected targets could be reliably identified as bats. The model analysis showed that the X-band marine radar was capable of detecting individual bat flights up to 1.0 km with > 70% probability of detection (POD). This suggests that bat echoes can be tracked with fewer non-detections, at least within 1.0 km of the radar. The findings of this study may also prove valuable for determining appropriate range settings for future radar bat studies. Specifically, when employing a tracking algorithm that permits discontinuities in target detection, it is crucial to select a survey range that can attain a higher POD for the successful establishment of a track (e.g., range with > 50% POD) [29,30,33].

The relatively high POD within the 1.0 km range was achieved by a combination of improved echo visibility and reduced ground clutter. We set the same color for all 15 levels that could be assigned to echoes. This setting improved the visibility of echoes, especially at lower signal intensity levels compared to the manufacturer's default settings (S1 Fig). Changing the color setting of the radar system may have increased echo detectability during manual tracking. As a note of caution, echo visibility enhancement also increases ground clutter and reduces the detectable

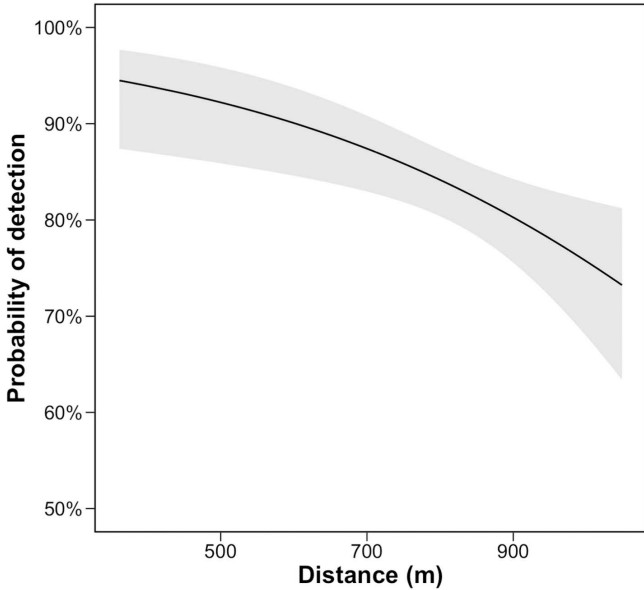

**Fig 5. Probability of bat echo detection in relation to distance from radar (mean±95% confidence interval).**

area on the radar image [23]. Changes in the color settings need to be applied simultaneously with ground clutter reduction techniques.

Distance exhibited a significant effect on POD, but the estimated effect size was small. This may be due to the limited sample size of the identified bat tracks (n = 25), which could lead to bias in the model estimates. Although a larger dataset is needed for robust modeling, it is worth noting that bat tracks consisted of a high proportion of detected echoes. The observed proportion of detection events per track ranged from 63% to 100% (mean = 83%), suggesting limited variation in POD. This may be the reason why the effect size of distance was estimated to be small. It should be noted that our results were obtained solely within this study's settings and observation range.

The maximum detection distance recorded in this study (1,048 m) was much greater than that of a single acoustic detector (approximately 40 m) for bats emitting low-frequency calls, as shown in previous literature [23,24]. This may represent an advantage of marine radar for monitoring bat movements over large areas. However, radar's horizontal detection range is highly dependent on landscape features that cause ground clutter. Furthermore, the detectable range for smaller species of bats is likely to be shorter. To clearly demonstrate the superiority of the radar, it is necessary to perform comparable validations in conjunction with acoustic devices or species of varying sizes.

Two factors require further verification. First, the flight altitude of bats could influence the detection capability of radar. In general, the signal power returned from a target decreases as the elevation angle from the beam center increases. In our study, the mean elevation angle from the radar position to the bat positions recorded by the ornithodolite was 3.1° (range = 0.9–8.7°). It can be assumed that bat echoes used in model analysis were detected within the main lobe (± 11° from the beam center) of the vertical beam axis, suggesting a relatively weak effect on detection probability. However, it should be noted that the radar detectability of bats flying at higher altitudes may not be consistent with the results of this study. This is due to the fact that we did not include elevation angle in the model analysis to avoid the issue of co-linearity. Second, variations in bat flight behavior may also affect radar detectability. For the purpose of accurate target identification, we focused our radar observations at dusk. Observed bats were in the transition phase after emerging from their roosts and exhibited straight flight paths. Therefore, we could not verify detectability of bats during foraging behavior, which involves more complex flight paths. It has been previously

documented that foraging bats exhibit erratic flight patterns [40–42]. The flight tortuosity of bats may influence successful detection and tracking, as has been suggested by previous avian radar studies [29,30]. At this time, the impact of behavioral irregularities on radar detectability remains uncertain. It is imperative to accumulate data on radar performance for behavioral variations both within and between bat species in future research endeavors.

We obtained radar tracks belonging to bats in-flight by matching them with ornithodolite tracking data. The ornithodolite demonstrated its capacity for providing a reliable ground truth for the radar performance test, enabling the direct identification of bats and achieving high accuracy in positioning. Although the number of radar tracks that matched the ornithodolite data was limited, we can be confident that the tracked echoes were bats. This conclusion is supported by the fact that the ranges of the speed and direction of movement of the identified bat tracks overlapped those of the ornithodolite data. During radar observations, most of the echoes were moving along paths from the viaduct, consistent with the observed behavior of the bats in the study area. Furthermore, almost all of the species observed visually in the area were Asian particolored bats. This evidence supports the assumption that the tracked radar targets were bats and not birds. An ornithodolite is an effective tool for collecting validation data, but direct observation at night remains challenging. Moreover, it was technically difficult to track the foraging behaviors of bats due to their erratic maneuvers. For these reasons, additional techniques are required to accurately identify species in complete darkness and track more complex behaviors.

### Ground clutter reduction

In this study, by lowering the antenna height, the visible area in the 1.5 km range was increased by a factor of 2.1. Reducing ground clutter made it easier to track bat echo movements on radar images. Our results suggest that the densely planted rice in the study area may have shielded the lower part of the vertical beam when the antenna height was lowered. A similar effect caused by natural objects, such as trees, has been mentioned in previous literature [16,19]. Prior studies have shown that dense foliage of vegetation scatters and attenuates a radar beam's radiated power as it propagates through the vegetation [43–45]. This reduces the amount of the reflected signal that returns to the radar receiver. The same process may have reduced the backscattering from the rice fields, thereby decreasing ground clutter.

Historically, previous radar studies have attempted to mitigate ground clutter through a variety of ways. These methods include the installation of clutter shielding fences [16,23] and the adjustment of antenna angles relative to the ground [24]. Additionally, dedicated avian radar systems have also implemented image processing algorithms that utilize raw reflectivity and/or velocity data to discriminate targets and ground clutter [46,47]. Although our adopted method has not been widely used, the effect of adjusting the antenna height on ground clutter in radar images has been demonstrated using a commercial marine radar [23]. For a portable commercial marine radar, adjusting the height of the antenna may be a simple option for achieving a larger detectable area.

Although ground clutter was successfully reduced in this study, we could not rule out the possibility that lowering the antenna height decreased radar coverage near the ground. This phenomenon has been shown by a simulation-based study that modeled radar vertical beam patterns with a clutter-shielding fence [23]. In that case, the received signal power from near the ground level (lower altitude) decreased with distance when the fence was installed. Since our study could not determine how rice field vegetation scattered and diffracted the radar beam, the issue of trade-offs between ground clutter reduction and detectable area is a matter for future verification. Furthermore, our performance evaluation was conducted at a single site with flat terrain. Given the highly site-specific nature of ground clutter within radar coverage, the methodology presented here should be subjected to testing at other locations to ascertain its applicability.

### Limitations

This study has potential limitations. The first shortcoming is the limited sample size of radar tracks that were successfully verified as bat flights (n = 25). It is important to note that the relatively high POD of bat echoes can be achieved within the

settings and observation range of this study. A larger dataset would be required for more robust analysis. The second limitation is its focus on a single location and a specific bat species. The detection capability of marine radar may be affected by differences in landscape features, the size of the target species, and the complexity of its behavior. In order to enhance the generalizability and applicability of our findings, it is necessary to validate the radar performance at multiple sites, in more complex terrain, and with other bat species of different size or flight behavior. To achieve this, we need to develop a technique for collecting ground truth data applicable to nighttime and complex behaviors.

## Conclusions

In this study, we verified the performance of a commercial X-band marine radar system for free-flying Asian particolored bats by analyzing the probability of echo detection in relation to observation distance. By employing an ornithodolite, we were able to acquire bat flight trajectories with exceptional positional precision, thereby eliminating the possibility of bird interference in our analysis. We demonstrated the survey distance necessary to achieve high bat echo detectability following the implementation of the ground clutter reduction technique. Our results further substantiate the bat detection capabilities of X-band marine radar, thereby contributing to the establishment of range settings that should be considered for detecting bat movements.

## Supporting information

**S1 Fig.  Echo color setting of the MDC-7910 marine radar system.**
(TIF)

**S2 Fig.  Relationship between elevation angle and distance.**
(TIF)

**S3 Fig.  Predicted probability curves of echo detection.** Model parameter estimates for each prediction curve were presented in S6 Table.
(TIF)

**S1 Table.  Results of clutter area calculation.**
(XLSX)

**S2 Table.  Tracking data obtained by ornithodolite.**
(XLSX)

**S3 Table.  Manual tracking data.**
(XLSX)

**S4 Table.  Manual tracking data for statistical analysis.**
(XLSX)

**S5 Table.  The relationship between bat echo detection and elevation angle.** A generalized linear mixed model (GLMM) consisted of echo detection as the dependent variable, elevation angle from radar as the explanatory variable and track ID as a random factor.
(XLSX)

**S6 Table.  Model parameter estimates change with a varying number of radar tracks.**
(XLSX)

**S1 File.  Results of post-hoc power analysis.**
(DOCX)

## Acknowledgments

We extend our sincere gratitude for the technical assistance provided by Yutaka Hagihara of Koden Electronics Co., Ltd. We also appreciate the invaluable insights and consultations offered by Hidehiko Tokushima of FRS Corporation Co., Ltd. particularly regarding radar installation techniques and clutter reduction methods. We would also like to thank Dr. Mark Brazil, Scientific Editing Services, for assistance in the preparation of the final draft of the manuscript. The data used for the map in Fig 1a have been provided by the High-Resolution Land Use and Land Cover Map (Japan) Version 25.04 of the Japan Aerospace Exploration Agency.

## Author contributions

**Conceptualization:** Takahiro Sato, Yoichi Kawaguchi.

**Data curation:** Takahiro Sato, Yoichi Kawaguchi.

**Formal analysis:** Takahiro Sato, Yoichi Kawaguchi.

**Funding acquisition:** Takahiro Sato, Yoichi Kawaguchi.

**Investigation:** Takahiro Sato, Yoichi Kawaguchi.

**Methodology:** Takahiro Sato, Yoichi Kawaguchi.

**Project administration:** Takahiro Sato, Yoichi Kawaguchi.

**Resources:** Takahiro Sato, Yoichi Kawaguchi.

**Software:** Takahiro Sato, Yoichi Kawaguchi.

**Supervision:** Takahiro Sato, Yoichi Kawaguchi.

**Validation:** Takahiro Sato, Yoichi Kawaguchi.

**Visualization:** Takahiro Sato, Yoichi Kawaguchi.

**Writing – original draft:** Takahiro Sato, Yoichi Kawaguchi.

**Writing – review & editing:** Takahiro Sato, Yoichi Kawaguchi.

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
