## [Decision Letter · Decision Letter 0]

11 Jun 2025

Dear Dr. Kawaguchi,

Thank you for submitting your manuscript to PLOS ONE. After careful consideration, we feel that it has merit but does not fully meet PLOS ONE’s publication criteria as it currently stands. Therefore, we invite you to submit a revised version of the manuscript that addresses the points raised during the review process.

We look forward to receiving your revised manuscript.

Kind regards,

Xuebo Zhang, Ph.D.

Academic Editor

PLOS ONE

Journal Requirements:

A. You may seek permission from the original copyright holder of Figure 1 to publish the content specifically under the CC BY 4.0 license. 

B. If you are unable to obtain permission from the original copyright holder to publish these figures under the CC BY 4.0 license or if the copyright holder’s requirements are incompatible with the CC BY 4.0 license, please either i) remove the figure or ii) supply a replacement figure that complies with the CC BY 4.0 license. Please check copyright information on all replacement figures and update the figure caption with source information. If applicable, please specify in the figure caption text when a figure is similar but not identical to the original image and is therefore for illustrative purposes only.

Reviewers' comments:

Reviewer's Responses to Questions

**Comments to the Author**

1. Is the manuscript technically sound, and do the data support the conclusions?

Reviewer #1: Yes

Reviewer #2: Yes

2. Has the statistical analysis been performed appropriately and rigorously?

Reviewer #1: No

Reviewer #2: Yes

3. Have the authors made all data underlying the findings in their manuscript fully available?

Reviewer #1: Yes

Reviewer #2: Yes

4. Is the manuscript presented in an intelligible fashion and written in standard English?

Reviewer #1: Yes

Reviewer #2: Yes

Reviewer #1: While the study presents an interesting application of marine radar for bat detection, the manuscript contains several critical limitations that compromise its scientific rigor and broader applicability. The methodology lacks robustness in key areas, the data analysis is insufficiently detailed, and the conclusions are overstated given the narrow scope of the research. The paper requires major revisions to address these shortcomings before it can be considered for publication.

1.The study was conducted at a single site with flat terrain and a specific bat species (Vespertilio sinensis), yet the conclusions imply broad applicability to "quantitative bat surveys over extensive spatial areas." The authors fail to acknowledge how radar performance may vary in complex landscapes (e.g., urban environments, mountainous terrain) or with species of different size/flight behavior. Without replicate experiments in diverse environments or comparisons with other bat species, the findings are inherently site-specific and cannot support the claimed generality.

2. The manuscript provides insufficient detail on how radar echoes were differentiated from non-bat targets (e.g., birds, insects, noise). While ornithodolite tracking was used to validate 25 tracks, the vast majority of radar echoes (386 tracks) were manually classified as "potential bat-like characteristics" without explicit criteria (e.g., velocity thresholds, flight patterns). This subjective approach introduces bias and undermines the reliability of the dataset. Additionally, the absence of machine learning-based classification or Doppler signature analysis—common in modern radar studies—limits the methodological rigor.

3. The analysis relies on only 25 validated bat tracks, which is statistically insufficient to robustly model detection probability as a function of distance. The generalized linear mixed model (GLMM) may lack power to detect subtle effects, and the high proportion of non-detection events (75 instances) could skew results. The authors do not report effect sizes (e.g., odds ratios) or conduct sensitivity analyses to assess how sample size impacts model stability. Without larger datasets or cross-validation, the claimed "70% detection probability at 1.0 km" is questionable.

4. The study focuses solely on distance as a predictor of detection probability but overlooks other critical factors. Bat flight altitude, for instance, could influence signal strength given the radar’s 22° vertical beam width, yet the authors do not analyze elevation angle data from the ornithodolite. Additionally, the manuscript fails to report environmental parameters (e.g., rain, fog) during data collection, despite known radar performance degradation in such conditions. Furthermore, the study only observes bats in straight-line post-roost flight, ignoring foraging behaviors like erratic maneuvers that may affect detectability. These omissions limit the model’s ecological relevance and its ability to generalize to real-world bat movement patterns.

5. The authors claim radar "circumvents range limitations" of acoustic surveys but do not directly compare radar detections with simultaneous acoustic data (e.g., bat pass rates) within the same spatial domain. Without such comparisons, it is impossible to quantify the added value of radar or validate its superiority in real-world scenarios. This missed opportunity weakens the study’s novelty and practical implications.

6. While lowering the antenna height reduced ground clutter, the authors do not theoretically justify this approach (e.g., using radar propagation models to explain how vegetation acts as a "clutter fence"). The analysis also fails to address trade-offs, such as reduced coverage at low antenna heights due to Earth’s curvature. A more rigorous discussion should link empirical results to radar theory and cite prior studies on clutter mitigation in avian radar.

Reviewer #2: This manuscript presents a well-designed, field-based study validating the performance of a commercial X-band marine radar (PONE-D-25-24846) for detecting free-flying Asian particolored bats (Vespertilio sinensis). The authors integrate ornithodolite-based tracking to match radar echoes with bat flight trajectories, analyze the impact of antenna height on clutter, and model the probability of detection (POD) using GLMM.

This study contributes novel, empirical evidence to a field lacking rigorous performance evaluations of radar systems for bat monitoring and offers practical guidance on range limitations, clutter reduction, and signal processing.

Reviewer comments:

Strengths:

Novelty and Relevance: Directly addresses a gap in validating marine radar detection capabilities for bats. High relevance for environmental impact assessments (EIA) and wind energy–wildlife conflict mitigation.

Methodological Rigor: Controlled validation using ornithodolite data for ground truth. Inclusion of clutter reduction, manual echo tracking, and statistical modeling (GLMM) for robust analysis.

Clarity and Detail: The manuscript is well-structured with detailed explanations of radar configuration, antenna adjustment, and data processing steps. Clear and appropriate use of figures (especially Fig 3 and Fig 5).

Practical Implications: Findings inform best practices for radar placement and system settings. Suggests radar applicability up to 1.0 km with >70% POD.

Major Points for Consideration and Improvement:

1. Add a paragraph in the Discussion explicitly acknowledging the limitations of the sample size and the need for multi-site validation.

2. Include explicit reporting of CI ranges in the Results text and briefly interpret the slope estimate of the distance variable in practical terms.

3. Adopt journal-standard reporting conventions for P-values (e.g., p < .001), include effect size estimates where possible, and always identify the test used (e.g., likelihood ratio test).

4. If no permit was needed for observational tracking, add a clarification (e.g., “No animal handling was conducted, so no ethics permit was required”).

5. Consider uploading S1–S4 Tables and corresponding radar images/trajectory files to a public repository for long-term access.

Minor Suggestions:

1. Title: Consider specifying the species (Vespertilio sinensis) in the title to emphasize the study’s biological focus.

2. Abstract: Mention the sample size of matched tracks (n = 25) to set expectations early.

3. Terminology: Use consistent terms for "ground clutter reduction" and "clutter fences" for clarity.

4. Typo Fixes: Line 186: "align with that of the rice fields" → consider rephrasing for clarity.

Recommendation:

✅ Accept with Minor Revisions

This is a well-executed, methodologically sound study with high practical utility and scientific value. The improvements suggested above would enhance clarity, reproducibility, and scientific rigor, but do not detract from the manuscript’s core contributions.

**Do you want your identity to be public for this peer review?** For information about this choice, including consent withdrawal, please see our Privacy Policy

Reviewer #1: No

Reviewer #2: No

---

## [Author Response · Author response to Decision Letter 1]

16 Sep 2025

Author responses to reviewer's comments

We greatly appreciate the reviewers taking the time and effort to provide constructive comments and helpful suggestions. Each suggested revision and comment brought forward by the reviewers was accurately incorporated and considered. We have carefully addressed all the reviewer's concerns. Please see below the author's point-by-point responses. All page numbers and line numbers refer to the revised manuscript file (Revised Manuscript with Track Changes) with tracked changes.

Reviewers' comments:

Reviewer's Responses to Questions

Comments to the Author

1. Is the manuscript technically sound, and do the data support the conclusions?

Reviewer #1: Yes

Reviewer #2: Yes

2. Has the statistical analysis been performed appropriately and rigorously?

Reviewer #1: No

Reviewer #2: Yes

3. Have the authors made all data underlying the findings in their manuscript fully available?

Reviewer #1: Yes

Reviewer #2: Yes

4. Is the manuscript presented in an intelligible fashion and written in standard English?

Reviewer #1: Yes

Reviewer #2: Yes

5. Review Comments to the Author

Reviewer #1

While the study presents an interesting application of marine radar for bat detection, the manuscript contains several critical limitations that compromise its scientific rigor and broader applicability. The methodology lacks robustness in key areas, the data analysis is insufficiently detailed, and the conclusions are overstated given the narrow scope of the research. The paper requires major revisions to address these shortcomings before it can be considered for publication.

Author response: We would like to thank you for your insightful comments. These comments have helped us to improve the quality of our paper. We are grateful for the time and energy you expended on our behalf. In the following sections, you will find our responses to each of your comments and suggestions. We hope our revisions address the concerns you have raised. All page numbers and line numbers refer to the revised manuscript file (Revised Manuscript with Track Changes) with tracked changes.

1. The study was conducted at a single site with flat terrain and a specific bat species (Vespertilio sinensis), yet the conclusions imply broad applicability to "quantitative bat surveys over extensive spatial areas." The authors fail to acknowledge how radar performance may vary in complex landscapes (e.g., urban environments, mountainous terrain) or with species of different size/flight behavior. Without replicate experiments in diverse environments or comparisons with other bat species, the findings are inherently site-specific and cannot support the claimed generality.

Author response: We appreciate the reviewer’s comment on this point. In accordance with your comments, we have removed the texts of the Conclusions section (p. 16, lines 393-395). To make the study’s limitations clearer, we have also added a new subsection titled “Limitations” to the Discussion (p. 16, lines 375-383). In the Limitations subsection, we have mentioned the need to verify factors affecting radar performance that have not been tested in this study including landscape features and interspecific differences in size or flight behavior. Additionally, for clarification, we have divided the Discussion section into three subsections: Bat detection performance (p. 12, lines 270-323), Ground clutter reduction (p. 14, lines 325-349), and Limitations (p. 16, lines 376-384).

2. The manuscript provides insufficient detail on how radar echoes were differentiated from non-bat targets (e.g., birds, insects, noise). While ornithodolite tracking was used to validate 25 tracks, the vast majority of radar echoes (386 tracks) were manually classified as "potential bat-like characteristics" without explicit criteria (e.g., velocity thresholds, flight patterns). This subjective approach introduces bias and undermines the reliability of the dataset. Additionally, the absence of machine learning-based classification or Doppler signature analysis—common in modern radar studies—limits the methodological rigor.

Author response: We appreciate the reviewer’s feedback regarding radar echo extraction procedures. In accordance with the reviewer’s comments, we have added further explanation for this point. While there is the possibility that the observed radar echoes may contain birds and insects, we believe most of the tracks extracted through manual tracking are from bats. To clarify, we have added the following texts to the Bat echo extraction subsection (p. 8-10, lines 184-228). The added texts read as follows:

[p. 8, lines 189-192] “Bird species observed in this area at dusk (grey heron, great white egret, and carrion crow) were generally scarce and only occasionally appeared within the radar observation area. In order to minimize contamination from bird movements, we focused only on echoes that emerged from the area around the locations of bat roosts (viaduct) and moved westward or southwestward. These were the major directions of the movement of bats observed in our study site.”

[p. 9, lines 203-213] “Furthermore, we examined whether the movement speed of the obtained radar tracks overlapped with that of bats rather than birds or insects. For the radar tracks, the mean ground-speed was 10.4 m/s (range = 4.2–16.5 m/s), suggesting that the majority of the tracked echoes were flying vertebrates since the typical flight speed of insects is less than 5 m/s [37, 38]. This data was consistent with the ground-speed of Asian particolored bats (mean = 9.9 m/s, range = 5.4–16.4 m/s) recorded by the ornithodolite. For birds observed in the same study area, the mean ground-speeds of grey herons, great white egrets, and carrion crows recorded by the ornithodolite were 25.0 m/s (range = 12.3–37.7 m/s, n = 13), 21.5 m/s (range = 10.6–37.9 m/s, n = 13), and 23.4 m/s (range = 14.4–30.2 m/s, n = 6), respectively. The ground-speed of the radar tracks was not significantly different from that of Asian particolored bats, but was significantly slower than that of birds (Steel-Dwass test: radar tracks vs. bats, W = −2.557, p = 0.348, radar tracks vs. grey herons, W = 8.441, p < 0.001, radar tracks vs. great white egrets, W = 7.943, p < 0.001, radar tracks vs. carrion crows, W = 5.880, p < 0.001). Therefore, we assumed that the manual tracking data were rarely contaminated by birds or insects.”

As the reviewer suggested, we agree that applying machine learning-based classification or Doppler signature analysis of radar echoes would be valuable. Regrettably, however, we are unable to do both because we lack sufficient training data for bats and birds, and the radar used in this study lacks a system for utilizing radial velocity. We are now collecting radar data of bats and birds for discriminating targets based on machine learning. Therefore, we would like to achieve this point in future research.

3. The analysis relies on only 25 validated bat tracks, which is statistically insufficient to robustly model detection probability as a function of distance. The generalized linear mixed model (GLMM) may lack power to detect subtle effects, and the high proportion of non-detection events (75 instances) could skew results. The authors do not report effect sizes (e.g., odds ratios) or conduct sensitivity analyses to assess how sample size impacts model stability. Without larger datasets or cross-validation, the claimed "70% detection probability at 1.0 km" is questionable.

Author response: We appreciate the reviewer’s comment. In accordance with your comment, we have added the odds ratio of the predictor (distance) to the Results section (p. 11, lines 257-258) and Table 2 (p. 11, lines 265-266). We have also revised the text describing the results of the analysis in the Results section as follows:

[p. 11, lines 256-258] “The distance showed a weak but statistically significant negative effect on the probability of detection (GLMM: odds ratio = 0.997, 95% confidence interval: 0.995−0.999, Table 2).”

Additionally, we have added the following text on the limitations of our sample size to the Discussion section:

[p. 12-13, lines 287-292] “Distance exhibited a significant effect on POD, but the estimated effect size was small. This may be due to the limited sample size of the identified bat tracks (n = 25), which could lead to bias in the model estimates. Although a larger dataset is needed for robust modeling, it is worth noting that bat tracks consisted of a high proportion of detected echoes. The observed proportion of detection events per track ranged from 63% to 100% (mean = 83%), suggesting limited variation in POD. This may be the reason why the effect size of distance was estimated to be small. It should be noted that our results were obtained solely within this study’s settings and observation range.”

Furthermore, in response to the reviewer’s concern, we have performed a power analysis using the R package simr (Green and MacLeod, 2016) with the current sample size although extracting the data on bat tracks had already been completed. The power of the predictor was 86.0% (95% CI: 83.7−88.1%), indicating an acceptable value compared to the traditional threshold of 80%. However, we did not include this in the manuscript because it was a post-hoc power based on results that had already been observed (did not indicate true power). It was not capable of providing new information beyond what was available in the model analysis results.

4. The study focuses solely on distance as a predictor of detection probability but overlooks other critical factors. Bat flight altitude, for instance, could influence signal strength given the radar’s 22° vertical beam width, yet the authors do not analyze elevation angle data from the ornithodolite. Additionally, the manuscript fails to report environmental parameters (e.g., rain, fog) during data collection, despite known radar performance degradation in such conditions. Furthermore, the study only observes bats in straight-line post-roost flight, ignoring foraging behaviors like erratic maneuvers that may affect detectability. These omissions limit the model’s ecological relevance and its ability to generalize to real-world bat movement patterns.

Author response 1 regarding effect of elevation angle: We appreciate the reviewer’s suggestion regarding this point. We agree that this is an important point. However, we believe it appropriate to use only distance in our model analysis for the following reasons. First, we have confirmed that there was a co-linearity between distance and elevation angle (Spearman’s rank correlation: r2 = −0.71, p < 0.001). To explain why we did not include elevation angle in the model analysis, we have added the following text to the Statistical analysis subsection:

[p. 10, lines 231-235] “The purpose of our analysis was to verify the effect of distance on the probability of bat echo detection. However, in addition to distance, bat flight altitude (elevation angle from antenna) also affects radar detectability. We checked for co-linearity between two variables: distance and elevation angle (recorded by the ornithodolite) from the radar position. Since a negative correlation was observed between distance and elevation angle (Spearman’s rank correlation: r2 = –0.71, p < 0.001), we only used distance in the following model analysis.”

Further, we have also analyzed the effect of elevation angle on bat echo detection using a simple generalized linear mixed model (GLMM) with elevation angle as the explanatory variable and track ID as the random factor. However, elevation angle showed no significant effect on probability of detection (GLMM: Odds ratio = 1.19, 95% CI: 0.89−1.60, z = 1.17, p = 0.24). For clarification, we have added the following text to the Discussion section regarding the effect of bat flight altitude on radar detectability:

[p. 13, lines 299-305] “Two factors require further verification. First, the flight altitude of bats could influence the detection capability of radar. In general, the signal power returned from a target decreases as the elevation angle from the beam center increases. In our study, the mean elevation angle from the radar position to the bat positions recorded by the ornithodolite was 3.1° (range = 0.9–8.7°). It can be assumed that bat echoes used in model analysis were detected within the main lobe (± 11° from the beam center) of the vertical beam axis, suggesting a relatively weak effect on detection probability. However, it should be noted that the radar detectability of bats flying at higher altitudes may not be consistent with the results of this study. This is due to the fact that we did not include elevation angle in the model analysis to avoid the issue of co-linearity.”

Author response 2 regarding conditions during data collection: Thank you for pointing this out. Our radar observation was conducted under conditions without rainfall or fog. We have added environmental conditions during data collection to the Methods section as follows:

[p. 7, lines 144] “Radar surveys were performed for a total of five days in early July 2023 under conditions with no rainfall or fog, both of which affect radar detection performance.”

Author response 3 regarding effect of bat behaviors: We appreciate the reviewer’s comment on this point. We agree that this point requires clarification. While we recognize that bat flight behaviors would affect radar detectability, tracking bat’s erratic movements by ornithodolite was technically difficult. For this reason, we focused our ornithodolite tracking efforts on flights soon after bats emerged from roosts at dusk. We have added the following text about the technical limitations of direct observation using the ornithodolite to the Discussion section:

[p. 14, lines 321-323] “Moreover, it was technically difficult to track the foraging behaviors of bats due to their erratic maneuvers. For these reasons, additional techniques are required to accurately identify species in complete darkness and track more complex behaviors.”

Additionally, we have added the following text regarding the effect of bat’s behaviors on radar detectability to the Discussion section:

[p. 13, lines 305-312] “Second, variations in bat flight behavior may also affect radar detectability. For the purpose of accurate target identification, we focused our radar observations at dusk. Observed bats were in the transition phase after emerging from their roosts and exhibited straight flight paths. Therefore, we could not verify detectability of bats during foraging behavior, which involves more complex flight paths. It has been previously documented that for aging bats exhibit erratic flight patterns [40, 41, 42]. Th

---

## [Decision Letter · Decision Letter 1]

6 Oct 2025

*Vespertilio sinensis*

Dear Dr. Kawaguchi,

Thank you for submitting your manuscript to PLOS ONE. After careful consideration, we feel that it has merit but does not fully meet PLOS ONE’s publication criteria as it currently stands. Therefore, we invite you to submit a revised version of the manuscript that addresses the points raised during the review process.

We look forward to receiving your revised manuscript.

Kind regards,

Xuebo Zhang, Ph.D.

Academic Editor

PLOS ONE

Journal Requirements:

Reviewers' comments:

Reviewer's Responses to Questions

**Comments to the Author**

Reviewer #1: All comments have been addressed

Reviewer #2: All comments have been addressed

2. Is the manuscript technically sound, and do the data support the conclusions?

Reviewer #1: Yes

Reviewer #2: Yes

3. Has the statistical analysis been performed appropriately and rigorously?

Reviewer #1: Yes

Reviewer #2: Yes

4. Have the authors made all data underlying the findings in their manuscript fully available?

Reviewer #1: Yes

Reviewer #2: Yes

5. Is the manuscript presented in an intelligible fashion and written in standard English?

Reviewer #1: Yes

Reviewer #2: Yes

Reviewer #1: The authors have responded promptly and thoroughly to each of the reviewers’ comments: they provided detailed explanations for key revisions, supplemented necessary supporting data where requested, and adjusted the manuscript structure to enhance readability, all of which demonstrate a strong commitment to improving the work’s academic quality.

As such, the manuscript, in its current form, is acceptable for publication.

Reviewer #2: Reviewer comments:

The revised manuscript demonstrates clear improvements in structure, transparency, and methodological explanation. The authors have carefully addressed many of the initial reviewer concerns by (1) introducing a dedicated Limitations subsection, (2) elaborating the radar echo extraction procedures, (3) clarifying statistical modeling decisions, (4) expanding discussion of clutter reduction, and (5) improving presentation and data accessibility.

The manuscript now reads clearly and is scientifically valuable as an empirical validation of a marine radar for bat detection. However, several key issues remain that require additional attention before final acceptance.

Major Comments:

1. Robustness of manual echo identification

Although the authors have added justification for manual selection based on flight direction and speed comparisons, the classification remains subjective. Only 25 out of 386 tracks were confirmed via ornithodolite matching, which may limit confidence in generalizing detection probabilities.

• Add a short sensitivity or validation analysis (e.g., varying inclusion thresholds for ground speed or track length) to show how detection probability estimates change under different selection criteria.

• If reanalysis is not feasible, state explicitly that the dataset’s small size and manual selection are limitations that may affect model generalizability.

2. GLMM model interpretation and reporting

The inclusion of odds ratios and 95% confidence intervals is appreciated, but practical interpretation is still limited.

• Report the odds ratio per 100 m increase in distance, or present a predicted probability curve (with 95% confidence bands) to illustrate detection decline with range.

• Include model diagnostics such as the random effect variance, sample size, and goodness-of-fit indices (AIC, R², or AUC).

• Consider including the post-hoc power analysis as supplementary material with a brief caveat about its interpretational limits.

3. Correlation and variable selection clarification

The manuscript mentions a “Spearman’s rank correlation: r² = –0.71,” which appears to be a notation error.

• Replace “r²” with Spearman’s ρ (rho) to avoid confusion.

• Provide the scatterplot of distance vs. elevation angle in the supplementary materials.

• Briefly discuss whether orthogonalization or residualization could allow assessment of altitude effects independent of distance.

Summary Recommendation

The manuscript has improved substantially and now meets most of the journal’s scientific standards. The remaining issues are primarily technical and analytical refinements to enhance transparency and interpretability.

Recommendation: Minor Revision (acceptable after clarification and supplementary additions).

**Do you want your identity to be public for this peer review?** For information about this choice, including consent withdrawal, please see our Privacy Policy

Reviewer #1: No

Reviewer #2: No

---

## [Author Response · Author response to Decision Letter 2]

29 Oct 2025

Manuscript ID: PONE-D-25-24846

Author responses to Reviewers

Dear Reviewers,

We would like to thank you for allowing us to resubmit a revised manuscript “Detection performance of an X-band marine radar system for free-flying Asian particolored bats (Vespertilio sinensis)” for publication in PLOS ONE. We appreciate the time and effort that the reviewers dedicated to providing comments on our paper. We have incorporated the comments and suggestions made by the reviewer. On the following sections, we present our point-by-point responses to the comments. All page numbers and line numbers refer to the revised manuscript file (Revised Manuscript with Track Changes) with tracked changes.

Reviewers' comments:

Reviewer's Responses to Questions

*Comments to the Author*

1. If the authors have adequately addressed your comments raised in a previous round of review and you feel that this manuscript is now

acceptable for publication, you may indicate that here to bypass the “Comments to the Author” section, enter your conflict of interest

statement in the “Confidential to Editor” section, and submit your "Accept" recommendation.

Reviewer #1: All comments have been addressed

Reviewer #2: All comments have been addressed

2. Is the manuscript technically sound, and do the data support the conclusions?

The manuscript must describe a technically sound piece of scientific research with data that supports the conclusions. Experiments must have

been conducted rigorously, with appropriate controls, replication, and sample sizes. The conclusions must be drawn appropriately based on the data presented.

Reviewer #1: Yes

Reviewer #2: Yes

3. Has the statistical analysis been performed appropriately and rigorously?

Reviewer #1: Yes

Reviewer #2: Yes

4. Have the authors made all data underlying the findings in their manuscript fully available?

The PLOS Data policy <http://track.editorialmanager.com/CL0/http:%2F%2Fwww.plosone.org%2Fstatic%2Fpolicies.action%23sharing/1/010f0199bc417245-27d08e82-c60e-47fc-bd01-9125d8deb722-000000/MKX_gcBesoc1dk9G0LJWgWc6aCvV-LXe9j8eNNNRvNc=231>

requires authors to make all data underlying the findings described in their manuscript fully available without restriction, with rare

exception (please refer to the Data Availability Statement in the manuscript PDF file). The data should be provided as part of the

manuscript or its supporting information, or deposited to a public repository. For example, in addition to summary statistics, the data

points behind means, medians and variance measures should be available. If there are restrictions on publicly sharing data—e.g. participant

privacy or use of data from a third party—those must be specified.

Reviewer #1: Yes

Reviewer #2: Yes

5. Is the manuscript presented in an intelligible fashion and written in standard English?

PLOS ONE does not copyedit accepted manuscripts, so the language in submitted articles must be clear, correct, and unambiguous. Any

typographical or grammatical errors should be corrected at revision, so please note any specific errors here.

Reviewer #1: Yes

Reviewer #2: Yes

6. Review Comments to the Author

Please use the space provided to explain your answers to the questions above. You may also include additional comments for the author,

including concerns about dual publication, research ethics, or publication ethics. (Please upload your review as an attachment if it

exceeds 20,000 characters)

Reviewer #1

The authors have responded promptly and thoroughly to each of the reviewers’ comments: they provided detailed explanations for key revisions, supplemented necessary supporting data where requested, and adjusted the manuscript structure to enhance readability, all of which demonstrate a strong commitment to improving the work’s academic quality. As such, the manuscript, in its current form, is acceptable for publication.

Author response: Thank you for accepting our revisions. Again, we would like to express our gratitude for your time and effort in improving our paper.

Reviewer #2

Reviewer comments:

The revised manuscript demonstrates clear improvements in structure, transparency, and methodological explanation. The authors have carefully addressed many of the initial reviewer concerns by (1) introducing a dedicated Limitations subsection, (2) elaborating the radar echo extraction procedures, (3) clarifying statistical modeling decisions, (4) expanding discussion of clutter reduction, and (5) improving presentation and data accessibility.

The manuscript now reads clearly and is scientifically valuable as an empirical validation of a marine radar for bat detection. However, several key issues remain that require additional attention before final acceptance.

Author response: We appreciate your acceptance of our manuscript revisions. We would also like to thank you for your comments and suggestions for improving our manuscript. In the following sections, you will find our responses to each of your comments. All page numbers and line numbers refer to the revised manuscript file (Revised Manuscript with Track Changes) with tracked changes.

Major Comments:

1. Robustness of manual echo identification

Although the authors have added justification for manual selection based on flight direction and speed comparisons, the classification remains subjective. Only 25 out of 386 tracks were confirmed via ornithodolite matching, which may limit confidence in generalizing detection probabilities.

• Add a short sensitivity or validation analysis (e.g., varying inclusion thresholds for ground speed or track length) to show how detection probability estimates change under different selection criteria.

• If reanalysis is not feasible, state explicitly that the dataset’s small size and manual selection are limitations that may affect model generalizability.

Author response: We appreciate the reviewer’s comments on this point. We agree that this is an important point. In accordance with the reviewer’s comments, we have conducted a validation analysis by varying inclusion thresholds for ground-speed. These thresholds were determined based on the ground-speed of bats recorded by ornithodolite. We have established the following thresholds to filter radar tracks: 1) minimum–maximum (n = 384), 2) 10–90 percentiles (n = 352), and 3) 25–75 percentiles (n = 218) of bat’s ground-speed. Using these data, we have modeled the effect of distance on the probability of echo detection. In addition, two models were included in the validation: one that included all radar tracks (n = 386), and one that included tracks via ornithodolite matching (n = 25). We have reported the results of this validation in the supplementary materials (S3 Fig and S6 Table), and added the following text in the Statistical analysis subsection. We hope that our additional analysis adequately addresses your concern.

[p. 10, lines 236-241] Additionally, we examined how detection probability estimates change with a varying number of included radar tracks. To do so, we modeled the probability of echo detection using subsets of radar tracks filtered by thresholds determined based on the bat’s ground-speed (S6 Table). The results showed that the effect of distance on detection probability follows a similar pattern across models (S3 Fig, S6 Table). Although the small sample size and manual selection of radar tracks may affect the generalizability of the model, for this study’s GLMM, we used radar tracks extracted by matching bat trajectories recorded by ornithodolite.

2. GLMM model interpretation and reporting

The inclusion of odds ratios and 95% confidence intervals is appreciated, but practical interpretation is still limited.

• Report the odds ratio per 100 m increase in distance, or present a predicted probability curve (with 95% confidence bands) to illustrate detection decline with range.

• Include model diagnostics such as the random effect variance, sample size, and goodness-of-fit indices (AIC, R², or AUC).

• Consider including the post-hoc power analysis as supplementary material with a brief caveat about its interpretational limits.

We appreciate your advice on this point. In accordance with the reviewer’s suggestions, we have addressed each comment as follows:

Author response 1: We believe this point has been addressed in Fig 5 (positions in the main text: p. 11, line 263 and p. 12, line 271), which illustrates a predicted probability curve over distance with 95% confidence interval. We hope that the figure addresses your comment.

Author response 2: We have included the random effect variance and sample size in Table 2 (p. 12), R2 and AIC in the text of the Results section (p. 11, line 260).

Author response 3: We agree with the reviewer’s comment and have included the result of the post-hoc power analysis in the supplementary material, along with text explaining its limitations (S1 File). We have also added the following text in the Statistical analysis subsection:

[p. 10, lines 234-236] “Due to the small number of identified bat tracks (n = 25), we checked the statistical power of the model with the current sample size by conducting a post-hoc power analysis (S1 File).”

3. Correlation and variable selection clarification

The manuscript mentions a “Spearman’s rank correlation: r² = –0.71,” which appears to be a notation error.

• Replace “r²” with Spearman’s ρ (rho) to avoid confusion.

• Provide the scatterplot of distance vs. elevation angle in the supplementary materials.

• Briefly discuss whether orthogonalization or residualization could allow assessment of altitude effects independent of distance.

Thank you for pointing this out. In accordance with the reviewer’s comments, we have made revisions as follows:

Author response 1: We have replaced “r2” with “Spearman’s rho” in the text (p. 10, line 227).

Author response 2: We have included the scatterplot of distance and elevation angle in the supplementary materials (S2 Fig, positions in the main text: p. 10, line 228).

Author response 3: We have added the following text in the Statistical analysis subsection. For clarification, we have also reported the result of a GLMM, which was analyzed the effect of elevation angle on detection probability in S5 Table.

[p. 10, lines 228-231] “While orthogonalization or residualization may allow simultaneous analysis for both variables, this approach was not used in this study. This is because it becomes difficult to interpret model estimates after applying these techniques. Additionally, elevation angle alone did not significantly affect the probability of echo detection (S5 Table).”

Summary Recommendation

The manuscript has improved substantially and now meets most of the journal’s scientific standards. The remaining issues are primarily technical and analytical refinements to enhance transparency and interpretability.

Recommendation: Minor Revision (acceptable after clarification and supplementary additions).

Again, thank you for giving us the opportunity to strengthen our manuscript with your comments and suggestions. We have worked hard to incorporate your comments and hope that these revisions are sufficient to make our submission suitable for publication in PLOS ONE.

Sincerely,

Yoichi Kawaguchi

Graduate School of Technology, Industrial, and Social Sciences, Tokushima University

2-1 Minamijosanjima-cho, Tokushima, Japan

TEL: +81-88-656-9025

E-mail: kawaguchi@ce.tokushima-u.ac.jp

Current address

Sado Island Center for Ecological Sustainability, Niigata University

1101-1 Niibo-katagami, Sado, Japan

TEL: +81-259-22-3885

E-mail: y.kawaguchi.sado@niigata-u.ac.jp

---

## [Decision Letter · Decision Letter 2]

10 Nov 2025

Detection performance of an X-band marine radar system for free-flying Asian particolored bats (*Vespertilio sinensis* )

PONE-D-25-24846R2

Dear Dr. Kawaguchi,

We’re pleased to inform you that your manuscript has been judged scientifically suitable for publication and will be formally accepted for publication once it meets all outstanding technical requirements.

Kind regards,

Xuebo Zhang, Ph.D.

Academic Editor

PLOS ONE

Additional Editor Comments (optional):

After careful checking, the authors just add some minor clarifications. Duet to this reason, this paper is suggested to be accepted after these clarifications.

Reviewers' comments:

Reviewer's Responses to Questions

**Comments to the Author**

Reviewer #1: All comments have been addressed

Reviewer #2: All comments have been addressed

2. Is the manuscript technically sound, and do the data support the conclusions?

Reviewer #1: Yes

Reviewer #2: Yes

3. Has the statistical analysis been performed appropriately and rigorously?

Reviewer #1: Yes

Reviewer #2: Yes

4. Have the authors made all data underlying the findings in their manuscript fully available?

Reviewer #1: Yes

Reviewer #2: Yes

5. Is the manuscript presented in an intelligible fashion and written in standard English?

Reviewer #1: Yes

Reviewer #2: Yes

Reviewer #1: The current form of the manuscript is fully acceptable for further processing. It now meets all the required academic standards, with a coherent framework, sound reasoning, and accurate citations that align with the journal’s guidelines. The authors’ proactive attitude toward revisions and their attention to detail have significantly enhanced the overall quality of the work, making it coherent, rigorous, and accessible to the target readership. We confirm that no additional revisions are needed at this stage, and the manuscript can proceed to the next phase of the publication process.

Reviewer #2: Reviewer Comments:

1. Robustness of Manual Echo Identification:

o Comment: The manual identification of radar echoes as bat flights remains subjective. Only 25 out of 386 tracks were verified using the ornithodolite, which may limit the generalizability of detection probabilities.

o Suggested Actions: The authors should conduct a sensitivity analysis by varying the inclusion thresholds for parameters like ground speed and track length to examine how detection probability estimates change under different criteria. If reanalysis is not feasible, they should explicitly mention the small sample size and manual selection as limitations that may affect model generalizability.

2. GLMM Model Interpretation and Reporting:

o Comment: The inclusion of odds ratios and 95% confidence intervals is useful, but the interpretation of the results could be more practical.

o Suggested Actions:

Report the odds ratio per 100 m increase in distance or provide a predicted probability curve (with 95% confidence bands) to illustrate detection decline with range.

Include model diagnostics such as random effect variance, sample size, and goodness-of-fit indices (AIC, R², or AUC).

Consider adding the post-hoc power analysis as supplementary material, with a caveat about its limits.

3. Correlation and Variable Selection Clarification:

o Comment: The manuscript mentions "Spearman’s rank correlation: r² = –0.71," which appears to be a notation error. The correct term is Spearman’s ρ (rho).

o Suggested Actions:

Replace "r²" with "Spearman’s ρ" to avoid confusion.

Provide the scatterplot of distance vs. elevation angle in supplementary materials.

Discuss whether orthogonalization or residualization could assess altitude effects independent of distance.

Summary Recommendation:

• Minor Revision: Acceptable after addressing the clarifications and statistical improvements.

**Do you want your identity to be public for this peer review?** For information about this choice, including consent withdrawal, please see our Privacy Policy

Reviewer #1: No

Reviewer #2: No

---

## [Editor Report · Acceptance letter]

PONE-D-25-24846R2

PLOS ONE

Dear Dr. Kawaguchi,

I'm pleased to inform you that your manuscript has been deemed suitable for publication in PLOS ONE. Congratulations! Your manuscript is now being handed over to our production team.

Kind regards,

on behalf of

Professor Xuebo Zhang

Academic Editor

PLOS ONE